# A New Concept of Associations between Gut Microbiota, Immunity and Central Nervous System for the Innovative Treatment of Neurodegenerative Disorders

**DOI:** 10.3390/metabo12111052

**Published:** 2022-11-01

**Authors:** Sayuri Yoshikawa, Kurumi Taniguchi, Haruka Sawamura, Yuka Ikeda, Ai Tsuji, Satoru Matsuda

**Affiliations:** Department of Food Science and Nutrition, Nara Women’s University, Kita-Uoya Nishimachi, Nara 630-8506, Japan

**Keywords:** gut microbiota, engram, neurodegenerative disorders, Alzheimer’s disease, Parkinson’s disease, amyotrophic lateral sclerosis, schizophrenia, inflammation, reactive oxygen species

## Abstract

Nerve cell death accounts for various neurodegenerative disorders, in which altered immunity to the integrated central nervous system (CNS) might have destructive consequences. This undesirable immune response often affects the progressive neurodegenerative diseases such as Alzheimer’s disease, Parkinson’s disease, schizophrenia and/or amyotrophic lateral sclerosis (ALS). It has been shown that commensal gut microbiota could influence the brain and/or several machineries of immune function. In other words, neurodegenerative disorders may be connected to the gut–brain–immune correlational system. The engrams in the brain could retain the information of a certain inflammation in the body which might be involved in the pathogenesis of neurodegenerative disorders. Tactics involving the use of probiotics and/or fecal microbiota transplantation (FMT) are now evolving as the most promising and/or valuable for the modification of the gut–brain–immune axis. More deliberation of this concept and the roles of gut microbiota would lead to the development of stupendous treatments for the prevention of, and/or therapeutics for, various intractable diseases including several neurodegenerative disorders.

## 1. Introduction

Neurodegenerative disorders are the most common factors of disability, which refer to the gradual decrease in function of the nerves of sensory, motor, and mental activity subsequent to the death of several neurons [1]. The nerve cell death accounts for the various neurological dysregulations of neurodegenerative disorders including Alzheimer’s disease, Parkinson’s disease, schizophrenia and amyotrophic lateral sclerosis (ALS) [2,3,4,5]. Similarly, some cases of autism and depression also result from nerve cell death [6,7,8]. Precise insight into the pathology of these diseases still remains elusive. Oxidative stress is defined as a condition of metabolic dysfunction facilitated by the discrepancy between the elevated production of reactive oxygen species (ROS) and the antioxidant defense activity in a body [9]. In one sense, a conceivable pathophysiology of neurodegenerative disorders might be recognized by the increased oxidative stress. For example, the elevated production of ROS has been hypothesized to play a key role in the development and poor outcome of schizophrenia patients [10]. Oxidative stress may also be increased in ALS patients, which may affect the mitochondrial dysfunction eventually leading to nerve cell damage and/or neuronal loss [11]. In particular, mitochondrial homeostasis is critical to maintain neuronal function and mitochondrial dysfunction is connected to neurodegeneration [12]. Neurons and glial cells are typically vulnerable to excess ROS because of comparatively insufficient antioxidant capabilities, which may increase vulnerability to neuronal damage and functional deficits [13]. It has been shown that these related mitochondrial disruptions of the oxidative pathways, several inflammatory cytokines, excess amounts of ROS, and altered microglia activities may bring harmful results to the process of nerve cell degeneration that eventually leads to nerve cell death [14] (Figure 1).

Increasing evidence indicates the complicated associations between gut microbiota, immunity and the central nervous system (CNS) [15]. Additionally, a variety of studies have shown the potential association between gut microbiota and neurodegenerative disorders including depression, autism, schizophrenia and Parkinson’s disease [16]. While regular gut microbiota could defend the CNS, the dysbiosis of microbiota might aggravate neurodegenerative and/or mental health disorders [17]. Hence, a good alteration of the microbiota could also support the inhibition and/or regulation of the development of neurodegenerative disorders. Although gut microbiota may play a critical role in the pathogenesis of ALS, for example, comprehensive studies implicating the intestinal changes in the pathology of neurodegenerative disorders are limited [18]. Recent studies could shed new light on the importance of disease-specific interactions between gut microbiota and neurodegenerative disorders [19]. Recently, we have suggested that immunological memory named “engrams” could restore the initial disease state in schizophrenia [20]. Based on this concept, innovative therapeutic strategies for several neurodegenerative disorders could be applied to the modification of gut microbiota. This review would emphasize the roles of the associations between gut microbiota, immunity and the central nervous system in the pathophysiology of neurodegenerative disorders, which could be modified by the alteration of gut microbiota as a hopeful treatment. This concept could also suggest supreme preventative and/or therapeutic strategies for the broader neurodegenerative disorders.

## 2. Inflammatory Neuro-Immune Responses

Inflammatory progression has a key role in various cellular processes and is suggested as the pathogenesis of neurodegenerative disorders [21]. Consistently, it has been revealed that neuro-inflammation triggered by bacterial or viral infections could induce schizophrenia in animal models [22]. In addition, it has been described as a reciprocal functional mechanism between the immune system and CNS [23]. For example, immune cells could modulate behavior and cognition of the host by direct interactions with the CNS [24]. A low-grade neuro-immune/inflammatory response is essential to keep the neurogenesis and/or the homeostasis of brain [5,7,25], suggesting that mild transient immune response might be employed as a restorative role in CNS. Consequently, an array of neuro-immune aberrations related to the chronic activated inflammatory reaction have been identified in patients with neurodegenerative disorders including schizophrenia [25]. Generally, an elevated level of inflammation markers in the blood and/or in cerebrospinal fluid (CSF) of the CNS has been detected in patients with neurodegenerative disorders [26]. Therefore, prospective treatment with anti-inflammatory medication has been suggested as a secondary treatment in patients with neurodegenerative disorders including schizophrenia or ALS [27]. It has been shown that extra prolonged stresses may be a robust risk factor for the development of some psychiatric diseases with a reduced number of mitochondria in the cortex [28]. 

Inflammatory oxidative stress may produce an excess amount of ROS which could be characterized as oxygen-comprising small molecules prone to react with several biological materials such as DNA [29]. In addition, an excess amount of ROS production could initiate an activation of autophagy in cells, suggesting an essential role for ROS in the activation of autophagy [30]. Generally, autophagy would play a protective role in cells; however, autophagy is also related to apoptotic cell death or necrosis in certain conditions. Additionally, autophagy could regulate the levels of several inflammations [31]. Hence, autophagy might be involved in the pathogenesis of neurodegenerative disorders. The significant effect of autophagy may be determined by the type of stimulus, cell types, the microenvironment, and/or other biological factors [32]. In intracellular signaling pathways, autophagy could be stimulated by the activated AMP-activated protein kinase (AMPK) during the situation of energy deficit in cells [33]. The activity of AMPK is also critical in the cells of the CNS for preserving neuronal integrity and for neuron survival against an excess amount of oxidative stresses [34]. Once activated, the consequently activated autophagy could overcome the inflammation by blocking the excretion of pro-inflammatory cytokines such as IL-1β and IL-18, which are an indispensable component of the autophagic mechanism responsible for the control of inflammatory immune response [35]. It is remarkable that damage in the neuro–immune interaction brings acute and/or chronic CNS pathologies, in which autophagy might be involved in neurons and/or glial cells [36].

## 3. Engrams and Neuro-Immune Responses in the Pathogenesis of Neurodegenerative Disorders

The CNS and the immune system might collaborate on various levels in a body; however, the mechanisms of holding the specific immune-challenge have remained vague. Very lately, it has been clearly shown that the brain keeps the facts of certain inflammation such as inflammatory bowel syndrome occurred in the body [37]. This specific memory seems to be an immunological remembrance called “engrams” [38]. Here, we would like to use this word “engrams” as the meaning of immunological remembrance *matching to the meaning of “memory-traces”.* The concept of engrams has been fairly hypothetical for the basic units of memory. Now, neuronal assemblies that hold the specific disease engrams have been known in the amygdala, hippocampus, and/or cortex, which may suggest that engrams are distributed among multiple brain regions functionally linking each other as an integrated engrams organization [39]. Associations of these engrams are thought to determine the situation of the host, either of health or disease, by engram arrangements, which may be frequently dependent on several environmental conditions [40]. Consequently, the immunological engrams could restore the initial inflammatory disease condition, if rebooted [38]. Created by stressful and/or repetitive inflammatory occasions, the engrams might commit to a slow progression of chronic diseases including neurodegenerative disorders [41]. Epigenetic changes such as DNA methylation or acetylation within the cells of the neuronal assemblies might be important mechanisms of the engram formation [42], which is also a significant factor for the fine-tuning of the function in the healthy brain [43]. Epigenetics may also stabilize engrams for the effective recovery of fear memory [44]. Therefore, engrams and/or epigenetic changes could be related to the immune consequences in the pathogenesis of various neurodegenerative disorders [45] (Figure 1). The synergistic arrangement of engrams might bring in the solid progression of several diseases, which involves the concept that any complex neurological and/or immunological consequences could result from the interaction of these engrams with immunity. Additionally, frequent subtle immune challenges might result in the stable formation of multiple engrams positioning independent information [46]. Synaptic variations might validate the specific development of engrams during memorizing for supporting memory. Maintenance of the memory might be achieved by a meta-plasticity mechanism that raises the change in neurons within an engram, which may be further encouraged by epigenetic regulators such as histone deacetylases (HDACs) [47]. In fact, the HDACs-related signaling pathways have been significantly associated with the alternative expression of several genes related to neurodegenerative disorders [48]. Consistently, some kinds of epigenetic regulators modified by environmental factors have been suggested as playing a crucial role in the pathogenesis of various neurodegenerative disorders [49]. In short, the brain could hold several specific inflammatory responses as information of pathological neuronal images called “engrams”. This concept could correctly elucidate the pathogenesis of various neurodegenerative disorders and the related CNS disorders, which might contribute to establishing a new strategy for the therapeutic interventions. 

## 4. How to Modulate the Engrams

Some engrams could potentially trigger and/or exacerbate the conditions of neurodegenerative disorders [50]. Therefore, clearing the bad memory of “engrams” might be favorable for the prevention and/or treatment of neurodegenerative disorders. In the experiment of dextran sulfate sodium-induced colitis, the authors applied the chemo-genetic procedure of the designer receptor exclusively activated by designer drugs (DREADD) system for the inhibition of engram activity [37]. However, it seems to be impossible to currently use this system in the clinical treatment of humans. Now, is it possible to clear the memory of engrams without neuronal cell death and/or any brain damage? This is the point for therapeutic interventions. In one possible way, synaptic removal could be achieved by microglia capable of initiating the oblivion of memories with engram cells [51]. It is considered that microglia can make synapse elimination a mechanism for forgetting memory retentions [51]. In addition, it has been reported that microglia are related to synapse density, learning, and/or memory [52]. There are significant associations between gut microbiota and demyelination by the microglia in the brain, suggesting that the crosstalk of gut-microbiota and brain-microglia might play a key role for the clearance of engrams [53]. It has been shown that regulation of the microbiota might be connected to the possible therapies of neurodegenerative disorders [54]. A gut–brain axis indicates a bidirectional connection between gut microbiota and brain, which is a vital assembly in the pathophysiology of several neurodegenerative disorders [55]. This concept might include the associations between gut microbiota and more broad CNS disorders. For example, it has been shown that the composition of gut microbiota might be associated with narcolepsy type 1 [56]. Changes in the conformation of gut microbiota may be accepted by the sympathetic vagal afferent nerve transmitting to the CNS via the microglial action, which in turn could produce and/or modulate the responses of engrams. Studies have proven that some species of bacteria could produce catecholamines and/or acetylcholine, which might contribute to the responses of the sympathetic nerve [57]. Some of vagal neurons in the sympathetic pathway usually have an afferent role for the microbiota-mediated adjustment of brain [58]. Convincing evidence has demonstrated the roles of gut microbiota in the pathogenesis of Alzheimer’s disease and/or Parkinson’s disease, which are partly mediated by modified microglial activity in the brain [59]. In fact, microglial dysfunction has been detected in a variety of neurodegenerative disorders including Alzheimer’s disease, Parkinson’s disease and/or ALS [59]. Possibly, the gut-microbiota–glia-brain–immune axis might be influenced by the production of inflammatory cytokines and/or by the reduction of favorable substances such as short-chain fatty acids (SCFAs), modifying the regulation of the sympathetic afferent nerve and glial cells [60]. For example, butyric acid, a key SCFA, might be connected with a favorable response in the treatment of schizophrenia, suggesting an important role in the gut microbiota–brain axis [61]. SCFAs can cross the blood–brain barrier (BBB) and could interact with microglia to regulate their functions [62]. Gut microbiota could also communicate with the brain through intricate communication systems, which incorporate the intestinal function with the cognitive and/or emotional brain via the neuro-immuno-endocrine mediators [63]. At least, some of the potential effectors in the gut could actually stimulate the sympathetic nerve pathway [58]. It has been demonstrated by reproducible and translatable findings that the efficacy of intervention could be achieved with microbial-derived metabolites for modulating the disease progression in ALS [64]. In addition, the impact of gut microbiota on brain function might be also related to brain cognition and/or perception. Therefore, several brain inflammations and/or neurodegeneration in the brain might be related to the action of the gut–brain axis [65], in which the immunity-linked processes might be associated with the neuronal responses to memory engrams [66]. Furthermore, there might be wide-ranging reciprocal connections between gut microbiota and immune-inflammatory responses with engrams, which have a critical significance in the function of the healthy brain and in the pathogenesis of various neurodegenerative disorders [67].

## 5. Utilization of Gut–Brain Axis for the Treatment of Neurodegenerative Disorders

The dynamic residency of microbes in the gut may play a fundamental role in managing host physiology. In addition, recent advances have emphasized the significance of gut microbiota in neurodevelopment with considerable associations with the onset and/or the progression of neurodegenerative disorders [68,69]. Furthermore, it has been shown that the dysbiosis of gut microbiota might worsen the symptoms of various neurodegenerative disorders [70]. Alterations in the composition of gut microbiota, termed gut dysbiosis, with an increased number of potentially pathological organisms might play a prominent role in the pathogenesis of CNS-related disorders. For example, ALS patients may often demonstrate some changes in their gut microbial communities compared to the paired healthy controls [71]. Furthermore, increasing gut dysbiosis has been shown to worsen the symptoms with ALS [72]. Evolving evidence also connects the gut dysbiosis to the exacerbation of impaired autophagy in the immune-mediated chronic neuroinflammation [73]. Interestingly, it has been reported that a pleiotropic drug modulating AMPK and/or autophagy signaling, such as metformin, could alter the gut microbiota and its metabolic processes [74]. Consequently, dietary approach to alter the gut microbiota could be advantageous for the treatment of neurodegenerative disorders [75]. Gut microbiota could regulate and/or inhibit the production of ROS to retain the host’s brain health [76]. It might be important to diminish the levels of ROS for neuroregeneration with neuronal stem cells [77,78]. In addition to the unfavorable effects for the stem cells, ROS might skew the function of microglia with the oxidized mitochondria in glial cells (Figure 1) [79]. Inflammatory factors, oxidative stress, and/or the alteration of microglia are known to limit neuroplasticity in the CNS [80]. In these ways, certain gut microbiota with the inhibition of ROS could probably prevent the incidence and/or attenuate the symptoms of neurodegenerative disorders by regulating the production of ROS and by clearing engram memory via the alteration of functional microglia in the brain (Figure 2). 

Innovative treatments for the neurodegenerative disorders including schizophrenia and/or Parkinson’s disease are progressing. Some methods for action that might efficiently influence the composition of gut microbiota may include fecal-microbiota transplantation (FMT) (Figure 2). By transferring the gut microbiota from a healthy donor, there have been promising signs of improving the capability of the gut microbiota for the treatment of neurodegenerative disorders [81]. In particular, the transplantation of microbiota containing *Faecalibacterium prausnitzii (F. prausnitzii)* could repair the structure of gut microbiota. For example, transplantation of F. prausnitzii has been utilized as an intervention method to treat dysbiosis of the gut microbiota connected to the inflammation preceding autoimmune diseases and/or diabetes [82]. In addition, it has been shown that patients with Parkinson’s disease have a considerably decreased number of *F. prausnitzii* compared to the control patients [83]. Moreover, the amount of *F. prausnitzii* may also work as a diagnostic and/or analytic biomarker for the successful procedure of FMT [84]. Consistently, the transplantation of fecal microbiota from patients with schizophrenia has triggered behavior alterations such as impaired learning and/or hyperactivity in the recipient animal [85]. Investigations with animal models suggest that the FMT is also valuable for the treatment of Parkinson’s disease [86]. Similarly, the administration of prebiotics and/or probiotics might be applicable to prevent and/or restore neurodegenerative disorders. The prebiotics are particular plant fibers which may stimulate the growth of healthy bacteria in the gut. The probiotics usually contain specific live organisms, which directly increase the populations of healthy microbes in the gut. Certain gut microbiota with prebiotics and/or probiotics have been shown to contribute to the treatment of ALS, suggesting that gut microbiota might be a new strategy for ALS treatment [87]. Furthermore, it has been shown that mild physical exercise has a cooperative effect on the gut microbiota with higher diversity [88], which might also improve the symptoms in schizophrenia and/or in major depression [89,90] (Figure 2). 

## 6. Next Perspectives

With no current cure for the various neurodegenerative disorders, therapeutics seem to have been concentrated on attempting to decelerate the progression of the disease and provide symptomatic treatments to maintain patient quality of life (QOL). Therapeutic exercise and/or rehabilitation are also recommended for patients to slow symptomatic progression [91]. Furthermore, multidisciplinary teams for therapy are known to improve patient QOL and prolong patient survival [92]. However, there is still no cure that could reverse the progression of these disorders. For example, at present, riluzole and edaravone may be two major disease-modifying drugs for the treatment of ALS [93,94]. The most broadly used drug, showing little beneficial effect on patient survival [95], riluzole, might have a complex mechanism of biochemical action [96]. Riluzole may prolong the survival of ALS patients by up to 20 months [97]. In the experimental study, enhanced mTOR levels and/or attenuated autophagic activity might have increased the survival of motor neurons, suggesting that the downregulation of autophagy might proffer a therapeutic procedure for the treatment of ALS [98]. Riluzole may show antioxidant capabilities against oxidative stress [99]. Another drug, edaravone, is also an antioxidant compound anticipated to reduce oxidative stress and remove lipid peroxidation [100]. Edaravone has been detected to have a therapeutic effect in ALS patients, exhibiting a decreased functional loss of several neurons [101]. Edaravone has been shown to remove hydroxyl radicals for the protection of neurons in ALS [102]. In addition, edaravone could also reduce excessive ROS, as a free radical scavenger, to prevent brain damage [103]. Interestingly, it has been shown that edaravone could ameliorate chronic stress-induced depressive symptoms in mice by regulating the gut microbiota [104]. The rather unsatisfactory efficacy of these conventional drugs might imply that new strategies are immediately needed to articulate therapeutic development for the treatment of ALS. The autophagic signaling pathway may be a crucial therapeutic target [105]. New therapeutic strategies for the ALS community are also mandatory in the struggle against an exponentially rising epidemiology of this disease [106]. 

Microbial fermentation-derived metabolites could introduce their effects via immunological and neuroendocrine mechanisms [107]. In particular, microbial neurochemicals such as amines, amino acids, and SCFAs, could contribute to the harmonious interactions between the intestinal microbial consortium, systemic immune cells, and the CNS [108], probably in part via the epigenetic mechanism. Notably, the gut microbiota–brain communication is bidirectional. This conversation might stabilize the physical and/or mental health condition, which could otherwise cause serious physical and/or mental health problems [109]. These psychobiotic treatments have exhibited favorable effects on neurodegenerative disorders by altering gut microbiota [110]. In addition, a probiotic supplement has been shown to amend the cognition of the recipients with Alzheimer’s disease [111]. It has been revealed that probiotics including *B. bifidum*, and/or *B. longum* supplementation in patients with Alzheimer’s disease could improve the cognitive function [112]. Therefore, the modulation of gut microbiota may be an encouraging therapeutic option to prevent Alzheimer’s disease [113]. Furthermore, such probiotics could inhibit many harmful effects of aging that are the recognized aggravators of various neurodegenerative disorders [114]. Interestingly, SCFAs generated by gut microbiota may improve the synaptic plasticity by reducing neuro-inflammation and epigenetically suppressing the accumulation of β-amyloid via the inhibition of HDACs in the mouse model of Alzheimer’s disease [115], which may be reminiscent of engram modulation via the interaction with microglia, as mentioned in Section 4. Psychobiotic treatments could be an encouraging strategy to improve the QOL for the patients who suffer from neurodegenerative disorders. With an intricate etiology and no current cure for many neurodegenerative disorders, broadening the understanding of the disease pathology is required to progress with patient care [116]. Through the modulation of functional pathways related to the brain–immune communication axis, the gut microbiota could influence the pathophysiology of neurodegenerative disorders. However, there is only sparse evidence on the precise role of the gut microbiota on the programming of immune cells in the underlying neurobiological pathways of neurodegenerative disorders. The microbiota metabolic pathways in the gut might be related to the secretion of inflammatory cytokines [117]. It is uncertain whether gut microbiota could decrease the risks of causing neurodegenerative disorders as a consequence of inhibiting the critical pathological processes. Comprehension of the precise relationship between gut microbial metabolic pathways and the clinical consequences would contribute a great deal to the progression of treatment for valuable interventions in neurodegenerative disorders. These tactics might be applicable for exploring the splendid function of gut microbiota. Therefore, prospective exploration is mandatory to understand the intricate interactions between brain engrams, certain immunity, and gut microbial communities. A systematized consideration of the roles of specific gut microbiota towards the development of various neurodegenerative disorders could confidently provide novel insight into the procedure of probiotics and/or FMT at least as a substitute approach for preventing and/or treating such diseases. It has been suggested that several neurodegenerative disorders have similar aspects to those of autoimmune diseases with the key pathogenic process mediating autoreactive T cells [118]. Biomarkers and possible therapeutic targets in neurodegenerative disorders may also overlap with those of several autoimmune diseases [119,120]. Hence, we here and now believe that the application of the gut–brain axis could expand for the superior treatment of autoimmune diseases and/or the related inflammatory diseases. Subsequently, forthcoming research should focus on the identification of disease-specific engram retention over time during the latent period of those diseases. The large number of researchers need to be united to comprehend the molecular mechanisms with better clarity to obtain superior therapeutic interventions for these intractable diseases. 

## 7. Conclusions

The essential role of the gut-microbiota–neuron–immunity interaction in the pathogenesis of neurodegenerative disorders such as Alzheimer’s disease, Parkinson’s disease, ALS, and schizophrenia has been shown here. In particular, that immunity can generally communicate with the engrams in the brain. Therefore, gut microbiota could provide support by taking favorable action via the modulation of engrams against the disease progression of several neurodegenerative disorders as well as probably several autoimmune diseases.

## Figures and Tables

**Figure 1 metabolites-12-01052-f001:**
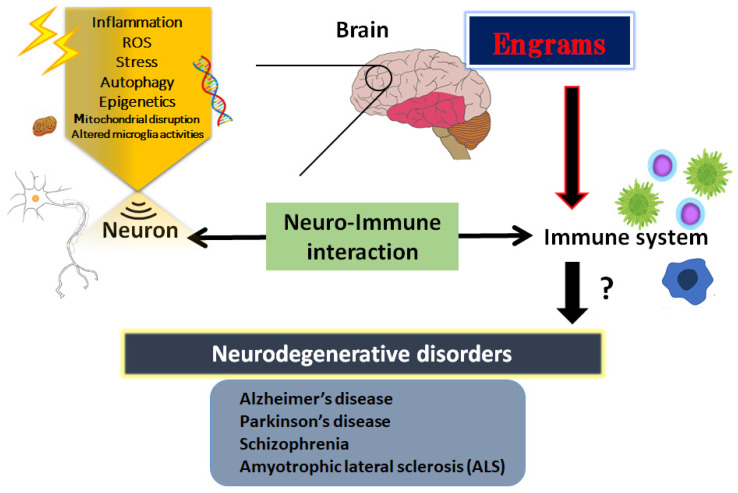
Schematic illustration shows an introduction to the essential role of neuro-immune interaction in the pathogenesis of neurodegenerative disorders such as Alzheimer’s disease, Parkinson’s disease, amyotrophic lateral sclerosis (ALS) and schizophrenia. The immunity could generally communicate with the brain or CNS. Illustration of the involvement for the pathogenic roles of various stresses, inflammation, ROS, epigenetics, and engrams is shown. Note that several significant items have been omitted for clarity. Abbreviation: CNS, central nervous system; ROS, reactive oxygen species.

**Figure 2 metabolites-12-01052-f002:**
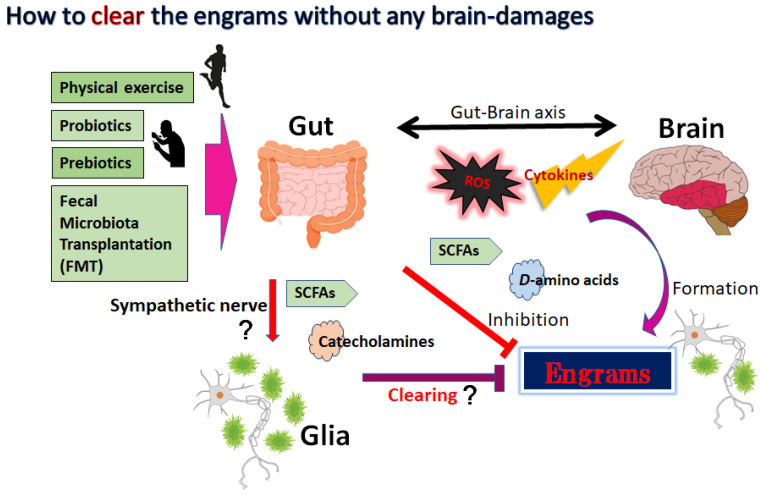
The gut microbiota could support favorable action against disease progression of neurodegenerative disorders by affecting the engrams and/or brain–immune axis, which may include the inhibition or production of cytokines, ROS, SCFAs, certain D-amino acids, and catecholamines. Mild physical exercise, probiotics, prebiotics, and fecal microbiota transplantation (FMT) might potentially be more successful than conventional symptomatic therapy for the treatment of neurodegenerative disorders. Arrowhead indicates stimulation whereas hammerhead shows inhibition. Note that several important activities such as cytokine induction or anti-inflammatory reaction have been omitted for clarity. Abbreviations: FMT, fecal microbiota transplantation; SCFAs, short-chain fatty acids; ROS, reactive oxygen species.

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
