# Peer review of "A New Concept of Associations between Gut Microbiota, Immunity and Central Nervous System for the Innovative Treatment of Neurodegenerative Disorders"

_metabolites, 2022, doi:10.3390/metabo12111052_

Round 1

Reviewer 1 Report

Dear Authors,

I read the perspective entitle: A new concept of associations between gut microbiota, immunity and central nervous system for the innovative treatment of neurodegenerative disorders”.

I found the paper very interesting, well written, quite clear and up to date.

The only request I have in order to make your work more clear is to introduce a table where you summarize the latest works that should indicate the animal model the type of neurological disease, the inflammatory response, the site where the engram seems to be stored, the intervention to counteract it and the effects of these interventions (please also include the references where you obtained these data). 

Best regards

Author Response

Reviewer 1

Comments and Suggestions for Authors

Dear Authors,

I read the perspective entitle: A new concept of associations between gut microbiota, immunity and central nervous system for the innovative treatment of neurodegenerative disorders”.

I found the paper very interesting, well written, quite clear and up to date.

The only request I have in order to make your work more clear is to introduce a table where you summarize the latest works that should indicate the animal model the type of neurological disease, the inflammatory response, the site where the engram seems to be stored, the intervention to counteract it and the effects of these interventions (please also include the references where you obtained these data). 

Best regards

Thank you very much for the good evaluation on our manuscript. As for the table suggested, we found more than hundreds of papers related to this issue. Summarizing them would be very tough at this point. As this is an important issue for the relevant research field, we would like to discuss and summarize them in the systematic review near future.

Reviewer 2 Report

In the present work, Yoshikawa et al. deepen on the complex picture of immune system, central nervous system (CNS) and gut microbiota in neurodegenerative diseases. The article is properly written and interesting. However i feel that some important details are missing, and i would recommend to the authors to review the following points: 

1) In section 4 "How to modulate the engrams" the authors start to focus on the role of gut microbiota in the interplay with the immune system and CNS. However i feel that addressing some important issues and a better structure related to this section will notably increase the quality of the manuscript. For instance:

     - Is there a specific footprint or certain microorganisms directly correlated with neurodegenerative disorders? 

      - Apart from SCFAs, are other microbial metabolites altered and playing a pathological role in neurodegenerative disorders (i.e. choline or tryptophan-derived metabolites, secondary bile acids, vitamins...)

     - Is there evidence of intestinal breakdown and bacterial translocation in these patients? If so, unraveling some crucial mechanisms such as LPS-induced neuroinflammation would aid to understand the intricate interactions between gut microbiota and immune system in the CNS.  

      - Perhaps, a simple figure of the MGB axis in the context of neurodegenerative disorders will aid to fully understand the manuscript.

2) In section 5 "Strategies modulating the MGB axis" some of the most relevant strategies targeting gut microbiota are presented. Despite FMT is a potential therapeutic approach, nowadays is not recommended for virtually any systemic or neurodegenerative disease and too much information is given in comparison to other strategies with more evidence. For instance: 

        - Despite the relevance of dietary interventions is mentioned, it should be important to give more details about it. What type of diet or group of foods would aid to patients with neurodegenerative disorders? What is the level of evidence?

       - Please, mention some type of prebiotic or probiotic/psychobiotic formulas which have been studied for patients with neurodegenerative disorders. Likewise the authors should describe, if available, some studies exploring the therapeutic use of postbiotics (i.e. SCFAs as epigenetic drugs) in models of neurodegenerative disorders

3) In the section 6 "Next perspectives" the authors talk about riluzole and edaravone. Do they have any effects on gut microbiota composition? 

Minor: 

Figure 1. Please, change "inframmation" for "inflammation"

Author Response

Reviewer 2

Comments and Suggestions for Authors

In the present work, Yoshikawa et al. deepen on the complex picture of immune system, central nervous system (CNS) and gut microbiota in neurodegenerative diseases. The article is properly written and interesting. However i feel that some important details are missing, and i would recommend to the authors to review the following points: 

  • In section 4 "How to modulate the engrams" the authors start to focus on the role of gut microbiota in the interplay with the immune system and CNS. However i feel that addressing some important issues and a better structure related to this section will notably increase the quality of the manuscript.

Thank you very much for the good assessment on our manuscript.

For instance:

     - Is there a specific footprint or certain microorganisms directly correlated with neurodegenerative disorders? 

We do not have found the paper that clearly shows the direct correlation between certain microorganism and a neurodegenerative disorder. Foot print can surely identify a certain microorganism. However, we believe that the direct correlation might be unnecessary for supporting the conclusion of this manuscript. Indirect effects even from vague collective bacteria would be enough to persuade this theory.

      - Apart from SCFAs, are other microbial metabolites altered and playing a pathological role in neurodegenerative disorders (i.e. choline or tryptophan-derived metabolites, secondary bile acids, vitamins...)

Thanks for the good comment. As for this issue, we have added some explanation in perspective section 6. At present, however, we think the other microbial metabolites, apart from SCFAs, could be faintly related to the formation of pathogenic engrams.

     - Is there evidence of intestinal breakdown and bacterial translocation in these patients? If so, unraveling some crucial mechanisms such as LPS-induced neuroinflammation would aid to understand the intricate interactions between gut microbiota and immune system in the CNS.  

True, this is a good point. Therefore, we have tried to hypothesize the mechanism via this engram-theory, by which the LPS-induced neuroinflammation could indirectly contribute to the pathogenesis of neurodegenerative disorders. We think the LPS-induced neuroinflammation may not the direct reason for the pathogenesis of neurodegenerative disorders. The LPS-induced neuroinflammation could bring a formation of certain engrams which might directly activate some of the immunity for the pathogenesis of neurodegenerative disorders.

      - Perhaps, a simple figure of the MGB axis in the context of neurodegenerative disorders will aid to fully understand the manuscript.

Absolutely. However, making a simple figure of the MGB axis in the context of neurodegenerative disorders is very difficult, confidently beyond our present capacity, because of the above intricate explanation.

2) In section 5 "Strategies modulating the MGB axis" some of the most relevant strategies targeting gut microbiota are presented. Despite FMT is a potential therapeutic approach, nowadays is not recommended for virtually any systemic or neurodegenerative disease and too much information is given in comparison to other strategies with more evidence. For instance: 

Current many failures even with concrete evidences could not deny the future success of conception.

        - Despite the relevance of dietary interventions is mentioned, it should be important to give more details about it. What type of diet or group of foods would aid to patients with neurodegenerative disorders? What is the level of evidence?

We speculate all might be probable. This manuscript is a “perspective” text. As you have noticed, we wrote this manuscript by the consideration of literatures in deductive inference for the development of this medical research field.

       - Please, mention some type of prebiotic or probiotic/psychobiotic formulas which have been studied for patients with neurodegenerative disorders. Likewise the authors should describe, if available, some studies exploring the therapeutic use of postbiotics (i.e. SCFAs as epigenetic drugs) in models of neurodegenerative disorders.

Nothing, at present sorry. We found no literature answering to this question. However, we believe such wonderful evidences should be explored in the upcoming intensive research. Here, we only indicate the direction of those researches for the innovative treatment of neurodegenerative disorders. 

3) In the section 6 "Next perspectives" the authors talk about riluzole and edaravone. Do they have any effects on gut microbiota composition? 

It has been shown edaravone could ameliorate chronic stress-induced depressive symptoms in mice by regulating gut microbiota [Appl Microbiol Biotechnol. 2021 Nov;105(21-22):8411-8426. ], which we have added in the text of section 6.

Minor: 

Figure 1. Please, change "inframmation" for "inflammation"

OK, thank you very much.

Reviewer 3 Report

This generally interesting publishable work should be supplemented with more references dealing with the microbiota's role in orchestrating the operation of the nervous and the immune system in health and disease as exemplified by the recent book:  Oleskin, A.V. and  Shenderov, B.A. (2020).. MICROBIAL COMMUNICATION AND MICROBIOTA-HOST INTERACTIONS: BIOMEDICAL, BIOTECHNOLOGICAL, AND BIOPOLITICAL IMPLICATIONS. Hauppauge (New York): Nova Science Publishers.

Besides, the language should be improved in terms of both grammar and vocabulary. 

Author Response

Reviewer 3

Comments and Suggestions for Authors

This generally interesting publishable work should be supplemented with more references dealing with the microbiota's role in orchestrating the operation of the nervous and the immune system in health and disease as exemplified by the recent book:  Oleskin, A.V. and  Shenderov, B.A. (2020).. MICROBIAL COMMUNICATION AND MICROBIOTA-HOST INTERACTIONS: BIOMEDICAL, BIOTECHNOLOGICAL, AND BIOPOLITICAL IMPLICATIONS. Hauppauge (New York): Nova Science Publishers.

Thank you very much for the good suggestion. According to this suggestion, we have added some explanation in the section 6 of future perspectives, citing the references by the same authors indicated. Those are as follows. Microbial fermentation-derived metabolites could produce their effects via immunologicaland neuroendocrine mechanisms [Microb Ecol Health Dis. 2016 Jul 5;27:30971.]. In particular, microbial neurochemicals such as amines, amino acids, and short-chain fatty acids (SCFAs), could contribute to the harmonious interactions between the intestinal microbial consortium, systemic immune cells, and CNS [Probiotics Antimicrob Proteins. 2017 Sep;9(3):215-234. ]. Notably, the gut microbiota-brain communication is bidirectional. This conversation might stabilize the physical and/or mental health state, which could alternatively cause serious physical and/or mental health problems [Probiotics Antimicrob Proteins. 2019 Dec;11(4):1071-1085. ]. 

Besides, the language should be improved in terms of both grammar and vocabulary. 

According to the suggestion, again we have gone over the text/abstract and amended typos and grammatical errors as much as possible to improve the manuscript more helpful to the readers.

Reviewer 4 Report

The authors present a new concept regarding the association of gut microbiota, immunity, and CNS, focusing on the innovative treatment of neurodegenerative disorders such as Alzheimer's disease, Parkinson's disease, and amyotrophic lateral sclerosis. The subject is interesting and discussed from the inflammatory neuro-immune responses to engrams and the role in neurodegenerative diseases and the possible future innovative treatments.

The paper needs a revision for the English language. 

Also, the images must be verified and corrected. It seems that Figure 2 brings nothing new to understanding the concept so it may be deleted. There should not be abbreviated words in the list of keywords.

At the end of the manuscript, the authors should include a section with conclusions.

Author Response

Reviewer 4

Comments and Suggestions for Authors

The authors present a new concept regarding the association of gut microbiota, immunity, and CNS, focusing on the innovative treatment of neurodegenerative disorders such as Alzheimer's disease, Parkinson's disease, and amyotrophic lateral sclerosis. The subject is interesting and discussed from the inflammatory neuro-immune responses to engrams and the role in neurodegenerative diseases and the possible future innovative treatments.

The paper needs a revision for the English language. 

According to this suggestion, we have gone over the text/abstract and amended typos and grammatical errors as much as possible to improve the manuscript more helpful to the readers.

Also, the images must be verified and corrected. It seems that Figure 2 brings nothing new to understanding the concept so it may be deleted. There should not be abbreviated words in the list of keywords.

According to this suggestion, we have deleted the Figure 2.

We have deleted the abbreviated words in the list of keywords, and replaced them with the term of full name.

At the end of the manuscript, the authors should include a section with conclusions.

According to this suggestion, we have added the conclusions in section 7.

Round 2

Reviewer 2 Report

The authors have addressed satisfactorily all my comments